# The Influence of Technologies in Increasing Transparency in Textile Supply Chains

Caterina Hauschild [1] and Angelica Coll [2,*]

1 Independent Researchers, 13347 Berlin, Germany
2 Chair of Logistics, Technische Univeristät Berlin, 10623 Berlin, Germany
* Correspondence: coll@logistik.tu-berlin.de; Tel.: +49-30-314-26745

**Abstract:** *Background:* In the current political discourse, supply chain transparency is seen as a key to improving the working and environmental conditions within textile supply chains. Additionally, the use of technology is increasingly being regarded as a means of reducing complexity and increasing transparency within these supply chains. While much research has been conducted to understand the impact of the textile industry on sustainability and the impact of technology on the overall performance of the textile supply chains, little attention has been placed on the following question: How do technologies affect transparency within the textile supply chains? *Methods:* We conducted seven interviews with actors from the textile industry. Based on these collected data, the relevance of selected technologies for improving transparency is established and the challenges of their implementation and impact on the industry are assessed. *Results:* Digital technologies, such as blockchain, the Internet of Things and dialog platforms, are promising instruments for transparency, even though their current implementation is not ideal. Furthermore, great skepticism on platforms for reporting (audits and complaint systems) is still prevalent. *Conclusions:* Since the influence of transparency on sustainability is conditioned by the goal orientation with which the technologies are implemented and used, we propose a framework for the implementation of the selected technologies that account for the interaction between said technologies in the textile supply chains.

**Keywords:** logistics; supply chain management; textile industry; digital technologies; digitalization; transparency

## 1. Introduction

Today, with 78 million workers along the supply chains, the textile industry is the second largest consumer goods industry in the world after the food industry [1,2]. Over the past decades, production in this industry has been outsourced to South-East Asian and Sub-Sahara African countries in an effort to remain profitable as fast fashion strategies reduce order volumes and product life cycles [3]. In response to this aggressive business model, globally fragmented and complex textile supply chains have developed [4]. As a result of these complex supply chains, the challenge for companies to track and trace their activities and generate transparency has increased [2].

At the same time, diverse stakeholders have been placing more pressure on companies in the textile industry to improve their sustainability, especially concerning human rights violations [5]. Nevertheless, "without better knowledge about the size of the industry and the scope of the problem [...], approaches designed to address these issues will not be able to solve the problem comprehensively" [6] (p. 45). Additionally, Straube et al. [7] cements the importance of transparency as the basis for supply chain sustainability when integrated into the corporate strategy. Therefore, technologies are being increasingly used by companies to facilitate data collection in textile supply chains, with which retailers hope to improve efficiency, product quality, sustainability, customer satisfaction and regulatory compliance [8]. From a business perspective, technologies are considered essential for

achieving greater visibility to identify and address challenges in complex supply chains, such as increasing data volume [9,10].

Technologies seem to be seen as a panacea for transparency and, at the same time, transparency as a panacea for sustainability. However, so far, it has not yet been studied, in a practical and industry-specific way, to what extent technologies have a positive influence on transparency in order to improve the sustainability of the textile industry.

Past research has been conducted on supply chain sustainability and its connection to transparency [2,11–13], on transparency and sustainability in the textile industry [1,5,14–18] or on the influence of technologies on transparency [9,10,19–23], partly connected to the textile industry [8,24–26]. Partly, technology use is associated with the sustainability of the textile industry, but often economic sustainability is the focus. Additionally, McGrath et al. [9] have investigated the roles of different types of technology on transparency.

In order to enrich previous knowledge, this research aims to investigate the impact of selected technologies on transparency for the purpose of improving social sustainability within the textile industry.

Therefore, the following research questions were formulated for this research:

- Which technologies are being used in the textile industry to increase transparency along supply chains?
- To which extent does the use of the respective technologies influence the increase in transparency along the supply chains of the textile industry?

## 2. Materials and Methods

The basis for the selection of technologies for this paper is found in the work of McGrath et al. [9], who examined across industries which technologies are being used in practice to increase sustainability visibility. They distinguish the effect of technologies on transparency according to collecting, processing and disseminating information. Since the focus of this paper is on the generation of information, or internal transparency, only technologies that collect information are included in the research scope, an overview of which technologies are analyzed in this paper.

This paper is based on qualitative research. In order to collect the necessary data to answer the research questions, we conducted semi-structured interviews following the criteria of Lamnek and Krell [27]. Since there may be differences of interest, especially between companies and NGOs, the two perspectives were the focus of the selection.

1. Textile and apparel companies that see themselves as sustainable and are responsible for the production itself and the possible implementation of technologies;
2. NGOs working for occupational safety and environmental protection in the textile industry. NGOs were chosen that advocate for workers and environmental conditions in the industry by publishing reports or campaigning, among other things. The advocacy group's point of view is particularly interesting, as they have direct channels to the workers, the companies as well as the government;
3. Service companies that help trading companies trace their supply chains;
4. Textile factories, as actors at the beginning of the supply chain, can assess the impact of technologies.

This work is limited to German-speaking contacts to avoid language barriers and the resulting scope for interpretation. Following this criteria, seven interviews were performed, as shown in Table 1. While the sample size is rather small considering the size of the textile industry worldwide, the insights obtained are still representative of the overall situation of technology implementation in the textile industry in Germany. The targeted interviewees were selected for their efforts in improving sustainability and transparency within the textile supply chains so that their expertise could be leveraged for the advancement of the textile industry in general.

**Table 1.** Composition of the sample; interviewees.

| Person ID | Role within the Institution | Actor in the Supply Chain | Institution ID |
|---|---|---|---|
| P1 | CEO/Founder | Blockchain as a service (BaaS) and sustainability platform provider for textile companies | NP1 |
| P2 | Consultant for sustainable supply chains and clothing | NGO | NGO2 |
| P3 | CSR Manager | Textile company | TBU3 |
| P4 | Auditor | Certification company | ZU4 |
| P5 | Sustainability Advocate & Consultant; former CSR Manager | Freelance; former textile company | TBU5 |
| P6 | Founder of the German NGO6; reg. coordinator | NGO | NGO6 |
| P7 | Technical coordinator | NGO | NGO6 |

The interview guideline consisted of the following three main blocks:

(i)   General questions about the organization's activities in the textile industry and especially in sustainability and transparency;

(ii)   Transparency in the supply chain, where the interviewees were asked about their understanding of transparency as well as the main challenges in the textile industry concerning transparency and its importance;

(iii)   Technologies for transparency, where the interviewees were asked about the relevance of the technologies for the industry, their role in generating transparency and the risks and challenges in their implementation.

Furthermore, the data collected in the interviews were evaluated with the help of qualitative content analysis, a widely used method designed by Mayring [28]. Firstly, the technologies mentioned by the interviewees were identified—during this step, a new technology category was created, namely, "Others". Within this category, we then proceeded to classify all technologies named by the interviewees that did not correspond to the initial set (i.e., the technologies shown in Figure 1). In the second step, the corresponding information provided by the interviewees was classified into a category (in this case: each technology mentioned). Subsequently, the indications concerning the impact of each technology were summarized and visualized in an impact matrix with the following dimensions: "transparency" and "information on sustainability". This allows for a clearer picture of the overall potential impact of the selected technologies.

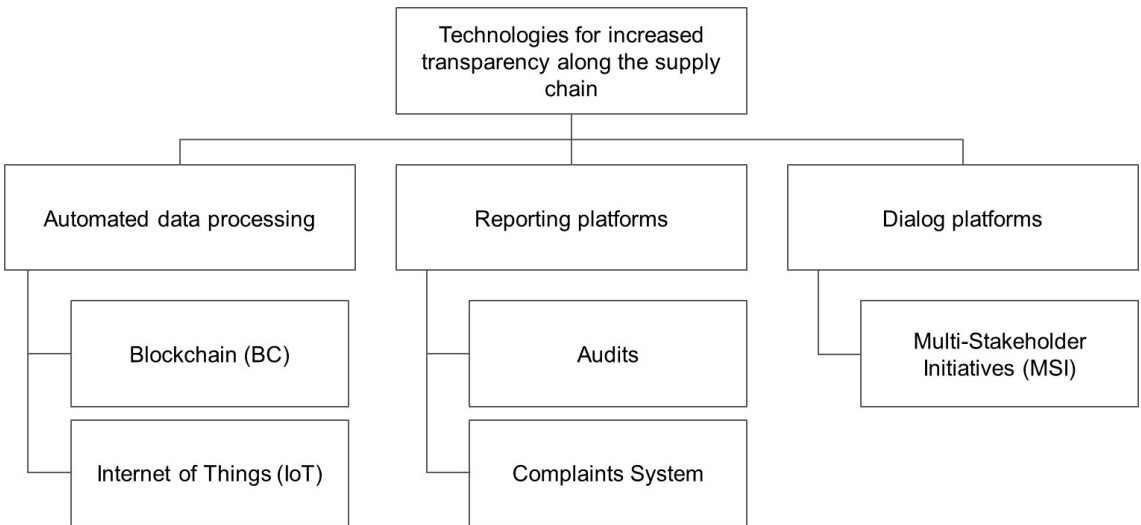

**Figure 1.** Overview of technologies for increased transparency along the supply chain.

## 3. Results

A detailed description of the information obtained within this research for each technology is given in this section. Table 2 shows the incidence of the technologies in the interviews.

**Table 2.** Breakdown of interviewees' statements on the technologies.

| Person. | IoT | BC | Audits | Complaint Systems | Dialog Platforms | Others |
|---|---|---|---|---|---|---|
| P1 | | x | x | | x | Sustainability Management Platform |
| P2 | | | x | x | | |
| P3 | | x | x | x | x | |
| P4 | | | x | x | | |
| P5 | | x | x | x | x | |
| P6 | | x | x | x | x | Unions |
| P7 | x | x | x | x | x | DNA-Analysis. Fine dust analysis |

### 3.1. Automated Data Processing

3.1.1. Internet of Things (IoT)

According to P7, the use of the IoT is significantly limited by the fact that the sensors and RFID tags cannot be installed at each stage of production but only after the manufacturing process of the fabric has been completed. Currently, according to P7, IoT for product tracking is only applied at the container level, i.e., during transportation. However, P7 sees great potential in the use of the IoT in earlier stages of the supply chain, especially in environmental monitoring. As an example, P7 cites the use of the IoT for monitoring air quality by measuring levels of toxins or temperature in factories. Currently, temperatures are measured annually via audits, but the IoT could collect data 24/7. For P7, the implementation is possible because it is not expensive, and the sensor can be set "so that it is not tampered with by the factory owner by hanging a cold rag over it", to lower the temperature.

3.1.2. Blockchain (BC)

Different assessments of the influence of BC emerge from the interviews. P1, P5 and P7 have dealt with the technology in more detail; P2, P3 and P6 have expressed their basic assessment of the technology.

In the interviews, BC was positively evaluated in that data can be collected, passed on securely and stored in a decentralized manner. BC creates trust, as the registered data can be assigned to the respective suppliers. In addition to information related to traceability, information on sustainability conditions can also be collected and evaluated. P3 and P5, who both work or have worked in textile companies, see great benefits in using BC. The data entered in the BC are based on the certificates previously issued by auditors. The use of the BC, in this case, is mainly for data preparation and disclosure of information via QR codes attached to the final products.

In contrast, P7 is critical, seeing no advantage in BC over a "quite banal database". However, the following example from TBU5 illustrates the advantage of using BC over simple databases, such as Excel lists: the supply chain of TBU5's products was nominated completely, from fiber origin to finished product (Nomination means that when a retail company awards a contract, it determines which suppliers its direct suppliers should purchase from), so it was assumed that the supply chain, including sustainability information, was fully known. After BC implementation, testing worked for 98 percent, but 2 percent had unknown fibers in the product. According to P5, they would not have found the error through their existing certification systems. However, P5 also mentions that BC cannot offer a 100% guarantee due to the dependence of the data quality and veracity on people. Nonetheless, both the system of Excel lists or on-site visits by the companies themselves and the cooperation with a certification system are prone to errors because the control instances behind them are not sufficient. Therefore, according to P5, BC's solution approaches are promising.

Several interviewees emphasize that BC technology as a whole is portrayed "in glowing descriptions" as being better than it actually is. It is seen as a "panacea" or "solver of everything". P7 is also "not convinced it's the best way to go, but it's just hip and makes money. [...] Some people are so enthusiastic about blockchain that they think it means information is always accurate." This interviewee remarks the positive attention BC receives and the trust it brings to the data as negative impacts of BC and is concerned with the energy consumption that the implementation of this technology requires. However, if BC is not public and does not go through the proof-of-work consensus mechanism, but only a few parties are allowed to write on BC, the energy problem would be solved and BC can be useful. According to P5, a verified life cycle assessment on BCs would be necessary to evaluate the energy consumption issue. Though, P5 questions whether a little more energy consumption, which may not be much when broken down to a garment, should take precedence over human rights security along supply chains. The interviewee criticizes that there is a lot of discussion around BC's $CO_2$ emissions rather than bringing change to the industry. Also, according to P1, the energy consumption of BC is only criticized because there is ignorance about how it works, which has neither a scaling nor an energy problem when applied to supply chains. P7, P6 and P3 express concern about tamper-proofness and trust and that immutability can lead to problems once the information entered is incorrect. False information can be entered consciously or unconsciously. According to P5, BC is more secure than the manual systems regarding deliberate misrepresentation. P1 explains a case that is an exception for tamper-proof storage of data as follows: In NP1, company IDs are entered into the system rather than the specific names of the suppliers. This aspect allows companies to exit the blockchain by registering another company for the respective ID. This mainly serves the privacy of the companies, as they must have the option under German or European law not to share their data. In addition, this circumvents the problem of incorrect data since the data can be changed via this detour. Nevertheless, P1 also believes that it is not through BC that the correctness of the entered data can be ensured, but through the logic of NP1's platform.

### 3.2. Platforms for Reporting

#### 3.2.1. Audits

As a criterion for transparency, the interviews asked about the assessment of the coverage of subcontractors through audits. P4 and P5 both explained that the ingredients (Ingredients are the individual components of the end products, some of which are bought in from subcontractors) of the products are nominated and can be traced well by means of the product catalogs. Products and all ingredients are certified in terms of traceability, mainly through transaction certificates, which can and should be verified. According to P4, when a product certificate is created, each actor in the supply chain must be checked for it. According to P5, this is a lot of work, which, nevertheless, "has to be done by everyone, because otherwise we can't sit down here and say: everything is fair and sustainable". Most of the time, certificates are not awarded after the first inspection but only after corrective actions have been taken. According to P4, when checking ingredients against the product catalog, there is little chance of not telling the auditors the truth. This is contrasted with P5's statement, as this actor has seen "more fake certificates than not fake". Depending on the standard the audit is guided by, audits differ in whether ingredient verification and subcontracts are covered. Since the standards leave room for interpretation, the claims and implementation of audits differ greatly between audit companies and between auditors. Audits are, according to P4, also situation- and operation-dependent. However, if audits are carried out conscientiously, they are a good system in their estimation. In the audit team of ZU4, so-called witness audits are carried out annually, in which a team-internal person accompanies the audit and checks whether the auditor is proceeding correctly.

P3 describes that supply chains can be fully traced due to the certifications given after verification through audits. Therefore, according to P3, audits are generally effective and furthermore a neutral procedure. When asked about the criticism that is raised against audits, P3 gives an example of discrepancies that have become known only after further examinations. P6 describes a similar situation but also points out that audits can have the following negative effects: worker interviews in a factory revealed that catastrophic conditions prevailed while the company's audits assessed the situation as being fine. "That makes it more difficult and structurally can't lead to truthful information".

Interviews with employees are an important part of the audits in order to obtain information about sustainability conditions that is as close to reality as possible. However, according to the interviewees, the implementation has shortcomings in several aspects. One criticism is that no offsite interviews are conducted, although they would be more likely to lead to truthful information. Workers in the factory would not disclose anything if there were possibly factory managers in the immediate vicinity who, for example, might threaten to fire the workers. Furthermore, the gender-parity composition of the audit team is relevant. P7 states that in a patriarchal environment, female workers would not respond to questions about gender-based violence. P2 mentions the check criterion on sexual assault from the PSCI reports as an example. An indication of one hundred percent compliance in the report means no sexual assault at all: "This is the indicator par excellence that no trust at all could be built to talk about such issues. It's not an indicator that the factories are so great, but rather that the factories are so bad." P4 doubts the statements of the workers, especially when the audit is announced. P7 implies that this is a main problem of a truthful audit. P4 disputes the criticism that only announced audits take place, stating that they conduct a certain percentage of unannounced audits per year. The coronavirus has exacerbated the problem of inspections in general and, specifically, the possibility of anonymous interviews.

Furthermore, the following structural weaknesses of the audit system were mentioned during the interviews: The long checklists are worked through under time pressure, which leads to gaps or errors. A two-day audit costs EUR 150: "That's spectacularly cheap. You can't do a real audit for that". The time required and the simultaneous pressure of time mean that neither more controls can be carried out nor can there be closer cooperation between the actors. According to P5, double-entry bookkeeping takes place in most

countries, e.g., working hours are documented and presented to auditors in a way that is different from the truth.

P2 mentions several requirements that need to change, as audits do not accurately reflect the situation at the audited sites. These include "cooperation with local actors", where certain standards have to be met, and contractual and financial independence between the auditor and the audited site. Moreover, it is important to look more closely at how the on-site inspections were carried out, and the results of each audit should "inform the reformulation or redesign of audits". According to P4, in order to fill information gaps and to have the opportunity as an auditor to take more effective action, a very different system rather than audits is needed. P7 says the following: "The best auditors would be [...] the workers. If they could mark the grievances without threat of consequences and also get time and opportunity to do so and the guarantee that it doesn't mean they will be dismissed, then those are the best auditors you could wish for".

When asked about possible solutions to certificate counterfeiting, P5 suggests the implementation of technologies. As concrete examples, though not yet scalable, she mentions BC, DNA analysis of fibers and materials or chip inlays. However, "Audits are still necessary to establish a status quo. [...] An audit is basically a status survey of a system. That means BC technologies also need to be regularly revised and audited by external entities, i.e., the system behind it".

### 3.2.2. Complaint Systems

According to P5, hotlines are "very important for the survey of social working conditions". P2 and P3 are also positive about hotlines, including as a useful complement to audits, as long as certain conditions are met. These include the local integration of the mechanism, especially considering language barriers and illiteracy, the level of knowledge on the topic on the part of the contact person, and that women are also included in the hotlines. P6 and P4 specify that the contact persons should not be auditors, but locally based NGOs, the ILO or trade unions. Anyway, it must be independent of the company so that workers have confidence. Even more effective would be "an anonymous complaint system within the company." In addition, there must be a protection for the reporters that guarantees the secrecy of their identity; otherwise, they could be exposed to repression. P7 describes the current situation of hotlines as follows: "Until now, there are almost only hotlines that are either made directly by factories, where it is unclear what happens with it, or it is said that it goes to the factory management. Of course, no one will call there". Another core factor for the current non-use is the workers' mistrust in the follow-up effect of the mechanisms. Interviews with workers by NGO6 concluded that the "biggest frustration was that it was completely unclear to them what would happen next".

FWF has a good process after P6 and P7. Nevertheless, according to P6, the entire system "must be structured differently. This is [...] not a question of technology", i.e., whether it is a hotline or something else. Central to this is the presence of trusted people on-site. Overall, it is a "complex thing that actually has to be set up in a social work way."

P6 presents the following scenario as effective: "Ideally, there would be a local office. In the production countries, partner networks would have to be established for NGOs or trade unions, which could then be the point of contact for complaints. [...] For example, Tierra eine Welt e.V. would be a local partner that would have competence and where workers would also go". Also, according to P4, an anonymous complaints office on-site is the best solution, just especially not the management of the company. A certain level of transparency is necessary in advance in order to find out at which company a complaint mechanism can be effective.

### 3.3. Dialog Platforms

Functioning as a learning space is crucial for MSIs to have a positive impact on transparency. According to P6, the fashion company Armed Angels, for example, is more knowledgeable about its own supply chain since becoming a member of FWF. However,

the effectiveness of an MSI as a learning space is highly dependent on which corporate strategy the member companies pursue.

NGO6 nevertheless withdraws from MSI because membership costs a lot of work but does not achieve much, even after years. The meeting of trading companies and trade unions takes place only very rarely and rather as a result of urgent actions. P7 criticizes that in many MSIs, contrary to the definition, only industries or brands are members, but not unions or the population. The only exception with real worker representation is FWF.

The main problems are the voluntary nature of MSI membership and the lack of sanctions for violating the codes of conduct. With a view to improving transparency and production conditions, there is no reason why firms should join MSI. It is more important to introduce effective laws and regulate labor inspections by the state. "All the bells and whistles with MSIs and audits is really just a stuffing box because this government institution doesn't exist [in producing countries]. Ideally, there should be one and minimum conditions should be laid down in law". Until now, there has been a lack of effective state institutions in production countries because the countries are unstable and, in some cases, very corrupt, lack the necessary financial resources and have to compete against each other on the world market. As an effective law, P7 cites the Uyghur Forced Labor Prevention Act in the U.S., which states that companies must prove that their goods imported from China were not produced with forced labor. P7 assumes that the EU will follow suit and that sustainability claims can no longer be made by companies if they cannot be proven.

The aspect of collaboration itself was discussed several times in the interviews, including MSI as a horizontal form. From the company's perspective, it is necessary to work closely with suppliers and establish a basis of trust in order to obtain the necessary information about the supply chain. Communication at eye level is seen by P1, P3 and P5 as the basis. With good cooperation, "you will also convince the supplier to work more transparently", as understanding and intrinsic motivation build up on the part of the suppliers. Moreover, it is important to be aware of possible language barriers and to make it as easy as possible for the suppliers.

According to P3, technologies that promote dialog are very important overall. A central aspect is also the exchange with network partners or other stakeholders about sustainability and the joint discussion of problems and search for solutions. However, communication costs time and money. Finding the right contacts also requires financial investment and is partly dependent on luck. Furthermore, according to P3, it is important to have been on-site in production locations as an entrepreneur to gain awareness about production conditions. P6 emphasizes that collaboration must not only take place between brand and factory management, but brands must also be in contact with workers.

*3.4. Other Technologies*

### 3.4.1. DNA and Fine Dust Analysis

DNA or fine dust analysis of fibers or materials can be used to determine their origin quite accurately, according to P7. For DNA analysis, actors in the supply chain cooperate by marking materials, such as organic cotton, with artificial DNA at the origin. The technology is cheap and ready for the market. If the actors in the supply chain do not cooperate, the fine dust analysis is a good alternative. Here, no artificial DNA marking has to be applied. By analyzing the dust, it is possible to say very precisely where materials come from. Even if the supply chain includes several countries, the origin and all the places where the material has been can be located with a 10 km radio accuracy.

### 3.4.2. Sustainability Management Platform

From P1's perspective, it is not BC but the logic of the *sustainability management platform* that is the technology that leads to transparency. Through the platform, suppliers enter their production and delivery data, risks are managed and actors along the supply chain can communicate through the platform. P1 compares the system to social networks such as Instagram, as the participating actors each have a profile that can be linked to the other

profiles. A network effect is created because the upload of a document is not only sent to one partner but can be seen by all actors linked to the profile. According to P1, the platform works as follows: data are collected from three dimensions. From the enterprise dimension, information about factories is obtained through audits and certificates, among others. The verification mechanism of audits uploaded to the platform is limited to checking the authenticity of documents. On the product dimension, documents about products and materials are collected. The supply chain dimension refers to the transparency of the chain: from a company perspective, information is generated about which suppliers are in the network. By creating an order in the platform, a brand can ask its known suppliers: "nominate through the system the suppliers, your suppliers, which you needed to fulfill this order". Through a step-by-step process, the supply chain can thus be better tracked and information on sustainability conditions can be requested. It is important that the platform considers possible language barriers and is intuitively designed so that suppliers are motivated to enter information. Mandatory training during supplier onboarding is crucial to ensure that suppliers share their information truthfully. In this context, it is also crucial to convey a sense of community to the suppliers so that, for example, information gaps are worked out jointly.

### 3.5. Transparency as Means to an End

According to P6, one problem is that companies often see transparency as the ultimate goal. "Transparency [however] is not an end in itself, but a means to an end". P5 describes it as follows: "Transparency and traceability is the basis for sustainability; the key to sustainability is which data and information is collected and, how they are evaluated." Also, according to P1, transparency is "the basis for efficient sustainability management". Data on production locations, as well as certificates through audits, are the basis for prioritization in sustainability management. Furthermore, several interviewees emphasized the importance of carrying out risk management after collecting information.

In addition, external transparency is stated as being indispensable. According to P7, internal transparency is useless without disclosure due to a lack of control. In contrast to common approaches that information is mainly disclosed to end consumers and stakeholders, according to P7, it should happen in a systemic way "so that workers, unions, scientists also have access to the data." P7 reasons, "One of the bigger problems is: when there are grievances, often it's unclear to workers what brand they actually worked for. Many can't read Western characters and simply don't know who they produced for and, therefore, where to go for compensation or some form of justice".

The interviewees point to further advantages but also to risks of more transparency, as follows: data protection must be considered when, for example, the addresses of farmers or home workers are published. They also include the safeguarding of interests by trade unions, whereupon disclosure can be viewed critically from a company perspective for reasons of competition. According to P3, the competitive risk arises from the fact that a trading company can be deprived of its suppliers by its competitors. Regarding the LkSG, companies like TBU3 have a competitive advantage if they can show several certificates about the supply chain. According to P5, "the competitive argument when it comes to transparency and traceability is [...] not valid." P5 justifies the statement by saying that the "taking away" of suppliers, which P3 mentions as a competitive risk, is not a realistic problem since it is time-consuming until a brand has established a production process.

"Anyone who, when it comes to traceability, transparency and sustainability, comes around the corner with the idea of competition, I don't think they should be producing products". According to P5, the problem with the textile industry is rather that many trading companies do not have in-depth knowledge of the industry itself.

### 3.6. The Schallmauer Effect

P6 explains a principle called the Schallmauer (sound barrier) effect. There is a sound barrier between trading companies and the upstream supply chain, which means that, for

example, solutions resulting from MSI conferences do not reach the actors along the supply chain and the effectiveness of trade unions and NGOs is severely limited. This aspect is often not considered in academic research "because most academic research deals with local stakeholders from a local perspective-and does not break through this "sound barrier" itself. It's also simply difficult and impossible to do without good networks in the countries where production takes place". P6, thus, highlights that the appropriate management system is the basis for companies to pursue subcontracting along the supply chain and penetrate the sound barrier. In addition, there is a disparity of power and influence from the trading companies with the highest influence on the home-based workers with the lowest. "Conversations about technology systems and management systems happen at the executive or middle management level". The narrative of retail companies, according to P5, is that they create jobs in the Global South and are therefore valued as good. However, according to P5, suppliers have a better insight into the complexity of networks, local structures and are better organized. Suppliers are also more open to technical innovations. In addition, with production planning on an equal footing, the competitive idea of external transparency would not be necessary. In addition, the workers have no influence or decision-making power. NGO6 "strongly plead[s] for workers and unions to be seen as full partners and not as disturbers of the peace".

Several interviewees mentioned that the information collected and passed on via the technologies always depends on the people who collect the information in the first place. Therefore, a certain degree of trust in the accuracy of the information is necessary. Core to trust, according to P3, is a long-standing working relationship with suppliers. With own visits on-site and a direct connection, trust can be built and transparency can be ensured. According to P1, intrinsic motivation must be created among suppliers, e.g., through an accurate onboarding of suppliers, to provide correct and detailed information.

## 4. Discussion

A key result of the interviews is the evaluation of the quality of information in the supply chain processes. Accordingly, it is not only important to examine which technologies are suitable for generating information, but also which technologies generate the right information. Figure 2 shows which technology can capture which information. The dark-colored technologies are those that were selected during the literature review for this work. The light coloring indicates the technologies that were presented initiatively in the interviews and for which no results from the literature were included. The results from the interviews confirm that traceability is the first step needed to collect further information about sustainability conditions along the supply chain and that both together create internal supply chain transparency.

### 4.1. Impact of Technologies on Traceability

As shown in Figure 2, digital technologies, i.e., IoT and BC, as well as DNA and fine dust analysis, are the focus of creating traceability. From the interviews, it has emerged that the IoT has a high potential to track the supply chain of the textile industry. Furthermore, DNA and fine dust analysis seem to be promising for tracing as a technique for checking audit certificates. A key difference is that DNA analysis requires supplier participation, so it could be subject to a similar limitation as RFID tags. With both methods, it is important to respect the privacy of the workers.

Different results are available on the assessment of BC. From the interviews, two narratives from practice stand out as significant. On the one hand, there is the example of TBU5, which received an error message for two percent of the fibers due to the implementation of BC, which would not have been detected via other systems. On the other hand, P1 highlights that BC only creates confidence in the data, while a sustainability platform is a technology that creates traceability in a practical context. When using BC, it seems crucial that it is connected to other systems. Essentially, there is agreement among the interviewees that BC leads to secure information through improved documentation and

automation. Regarding the aspect of traceability, the potential is thus seen in BC through tracking and tracing.

**Figure 2.** Technologies and their possibilities to capture information.

Both complaint systems and audits are primarily conducted at locations that are already known, with the aim of recording conditions on-site. Therefore, they are not directly associated with their contribution to traceability. One aspect that was highlighted during the interviews is the possibility to check where the ingredients, i.e., inputs, come from using the product catalogs and transaction certificates during audits. It is clear from the interview with P2 that many subcontractors are not recorded, and the certificates are criticized for not being sufficient [26]. Thus, it is questionable to what extent only the direct suppliers are included in a product certificate and not the subcontractors. Additionally, audits, in the same way as RFID tags, could cause a break in the flow of information.

The complexity of the supply chain is largely caused by the involvement of intermediaries and subcontractors, which is particularly common in the textile industry [2,6,29]. Reducing these instances is one way to improve traceability. In interviews, product nomination was presented as a way to bypass them. If the corporate strategy is focused on sustainability, e.g., explicitly through the integration of a sustainable supply chain management, it is assumed that the extra effort to nominate products is a realistic step to establish traceability.

*4.2. Influence of Technologies on the Information on Sustainability*

It was possible within this research to identify a trend in which technologies can capture information on sustainability and whether differences exist between capturing environmental and social conditions. The challenges to capture have been found to be the conscientiousness of people and the organizational structures of the textile industry.

According to P5, the information on sustainability can be easily entered into the BC. The fact that the data are entered directly by the suppliers can be seen as an opportunity but also as a challenge. On the one hand, this indirectly creates an exchange along the supply chain, and the suppliers can act self-determined. On the other hand, the information must either be trusted, or audits are still necessary to control the situation on-site.

Since the scope of audits depends on the standard according to which audits are carried out, the influence of audits on increasing transparency depends on the standard. It can be evaluated positively that the audits monitor the situation on-site as comprehensively as possible using the checklists. However, the way it is carried out leads to information gaps, mainly in the following three areas: Firstly, not all information is shared with trading companies as reports are aggregated [30]. Secondly, the information about social conditions is sometimes not entrusted to the auditors by the workers. And, if information is given, it cannot be included in the audit report without verification through documents. Thirdly, certain aspects of the checklist, such as the inspection of buildings, cannot be properly performed by auditors. Because audits can therefore have large information gaps, but certificates are nevertheless issued, which companies refer to, e.g., when accidents occur in factories [31], the influence of audits on information on sustainability is sometimes even classified as negative.

In the interviews, it emerged that the complaint systems tend not to be affiliated with the companies themselves, but they were increasingly addressed as part of the audits or as an affiliation with an MSI (Defined by the OECD as a "multi-stakeholder grievance mechanism"). Overall, they are found to be very relevant for capturing information on sustainability at the social dimension. Like complaint systems, unions could also tend to represent workers' voices. Specifics of unionization were not explored further within this research.

*4.3. Authenticity of Information for Transparency*

Within this research, it was found that the authenticity of the data is a major challenge in the generation of information and has an impact on the effectiveness of transparency. This gives importance to looking at which technology can best generate the right data.

In this work, authenticity is attributed primarily to the dependence on the people who capture and transmit the information. It is reasonable to assume that technologies that function independently of humans capture more truthful information. Figure 3 depicts a scale on which the assessment of each technology is plotted. On the right-hand side, the technologies to which the highest information authenticity is assigned are shown.

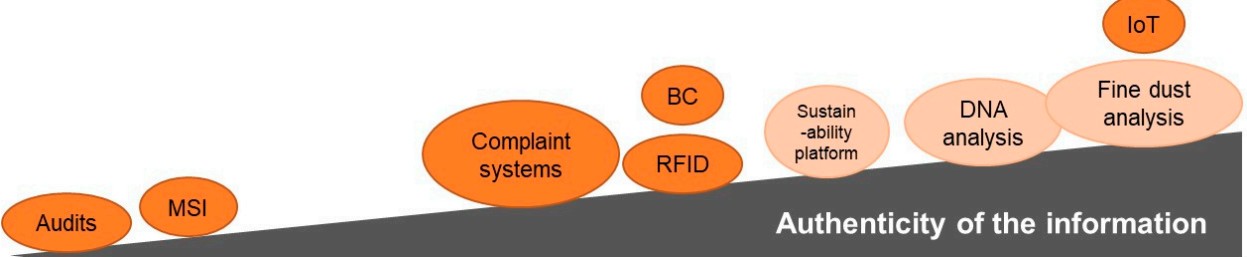

**Figure 3.** Classification of technologies with regard to the authenticity of their information.

IoT technologies or especially fine dust analysis, are credited with being able to collect data independently of suppliers [9]. In connection with BC, the concept of trust in the information is at the top of the list. However, RFID tags can be easily falsified [32], and the information that is entered into a BC is at the discretion of the network participants and is based, among other things, on the audit results. Deliberate misstatements are nevertheless more likely to occur with manual systems than with the use of BCs. The overall tendency is that digital technologies are attributed more authenticity than audits, complaint systems and MSIs.

The authenticity of the information provided during audits is, to some extent, strongly criticized. Neither is the information provided by the auditors trusted, nor do the auditors trust all the data they receive from the factories about the conditions on-site. The fact that audits are relatively heavily criticized may also be due to the fact that they have been used

for longer and are more deeply integrated into the processes of the textile industry than the newer technologies such as IoT and BC.

Complaint systems are considered to have a high potential, as the information comes directly from the workers and is therefore not distorted via intermediate stages. It is crucial that neither auditors nor factory managers are the contact persons, but independent, locally based entities such as trade unions or NGOs so that the workers can build trust and provide real information. As there are currently few grievance systems in place that workers trust, they are ranked rather than left on the scale, with the comment that there is much potential for improvement.

### 4.4. Implementation of Technologies in the Textile Industry

Table 3 shows how the technologies described affect the two aspects of transparency, traceability (here T) and information on sustainability (here IoS), from the point of view of the interviewees. It thus provides an overview of which technologies are classified as implementable or worthy of implementation.

**Table 3.** Overview of the interviewees' assessment of the technologies.

| | IoT | | BC | | Audits | | Complaint System | | MSI | |
|---|---|---|---|---|---|---|---|---|---|---|
| | T | IoS | T | IoS | T | IoS | T | IoS | T | IoS |
| P1 | | | Reference to sustainability platform | | | (light green) | | | | Cooperation | |
| P2 | | | | | | (light green) | (magenta) | (light green) | | |
| P3 | | | (dark green) | | | (blue) | (magenta) | | Cooperation | |
| P4 | | | | | (dark green) | (blue) | (magenta) | | | |
| P5 | | | (dark green) | | | (light green) | (magenta) | | Cooperation | |
| P6 | | | (blue) | | (magenta) | | | (light green) | MSI | |
| P7 | (dark green) | (light green) | (magenta) | | | | | (light green) | MSI | |

Legend: Dark green—positive effect. Magenta—no or negative effect. Blue positive and negative aspects mentioned or undecided. Light green—currently no effect with growth potential. Empty—No comment.

Since IoT for traceability in the textile industry is seen to have high development potential and economic benefits for companies, it can be assumed that the technology will continue to develop and find increased applications. The use of the technology appears to make sense for real-time detection, primarily from the fabric manufacturing stage onwards.

Opinions about BC vary widely. On the one hand, it is seen as a forward-looking and overall suitable technology; on the other hand, its use is seen as redundant and the focus on technology as negative because the risks of the textile industry should be solved differently. In addition, it is doubted whether some companies implement BC primarily because they are aligned with the competition, although it is not aligned with their business risks [33]. The interviewees who work in NGOs, i.e., P2, P6 and P7, are rather negative towards BC and do not see it as purposeful (Table 3), whereas it is seen as having great potential in the current scientific discourse.

However, the use of audits is the most widespread technology in the textile industry and their effectiveness is much debated, with opinions differing between actors [9,18]. The current audit systems' effectiveness is fundamentally questioned by several interviewees. It is assumed that, in the future, there will be an increased focus on the use of dialog-promoting technologies instead of controls through audits.

No potential is seen in complaint systems for traceability, but all the greater for information on sustainability. All interviewees see great potential in the technology for the textile industry as long as certain requirements are given. In addition to the necessary

confidentiality of the systems, educating workers on how to manage the complaint system is fundamental to its use. Without these aspects, workers would not use a complaint system, and implementation would be redundant.

MSIs are already represented in the industry with different orientations. Since several NGOs and interviewees highlighted the FWF as a positive example [34,35], the recommendation is to align complaint systems and the structure of MSIs with the FWF's mode of operation. In addition, MSIs should be legally binding.

The Schallmauer effect is particularly interesting because it fundamentally questions both the system of how transparency is created and the individual technologies that are either already widely used for this purpose (especially audits) or are now increasingly being used (IoT and BC). In order to create comprehensive transparency that incorporates the perspective of workers and, based on this, identifies proposed solutions for the risks of the textile industry, the inclusion of the Schallmauer effect is essential.

## 5. Conclusions

Regarding the extent to which the use of a respective technology influences transparency, it was found that several technologies can capture partial aspects of traceability or information on sustainability. Nevertheless, the interaction between technologies is necessary to create overall transparency along the textile supply chains.

The way the technologies work differs fundamentally. While MSIs and sustainability management platforms, for example, tend to indirectly lead to companies collecting data, DNA and fine dust analyses are more meaningful for traceability, IoT for ecological conditions or complaint systems for social conditions. It, therefore, seems of little use to evaluate the technologies alone. Here, we present a proposal for the textile industry that builds on the strengths of the technologies addressed in this paper and considers the risks of the textile industry.

First, DNA and fine dust analyses should be used to determine the origin of the products. Especially with these two technologies, it is important to treat the data confidentially and not to create any disadvantages for workers, e.g., by making home workplaces known. In cooperation with suppliers, RFID tags can be used as a technology of the IoT, as they are seen as having the potential to record the flow of goods in real-time. In this way, trading companies can identify possible production difficulties or routes via intermediaries. In addition, IoT techniques can measure conditions at production sites, such as temperature. If possible, risks are identified, real-time monitoring could be used to contact suppliers directly when the problem arises and to enter into a dialog.

Combining a sustainability management platform with the use of BC seems promising to ensure secure documentation of data along the supply chain and, at the same time, to promote exchange between actors. By using BC, for example, companies can more easily find errors in data along supply chains, which are complex, especially in the textile industry.

One approach to improving the impact of audits would be for the auditors to work in teams that include people with different expertise, for example, to check the building statics and to be able to hold sensitive discussions with the workers. There should be enough time for the auditors and the possibility to build up a relationship of trust with the workers. However, in the case of financing by the textile companies, conflicts of interest still cannot be ruled out. It would also be conceivable for auditors to collect information along the entire supply chain based on the information in the transaction certificates. However, from the information obtained directly from auditors for this paper, it can be concluded that this is not currently performed. Audits at least lead to the fact that companies create documentation and thus remain aware of the tasks.

More promising are complaint systems, which should be implemented by each production site to provide a space for workers to share information directly.

Dialog-enabling technologies are the foundation on which all technologies can build. Recognition of suppliers and workers as the ones who make the supply chain work is important for effective dialog. Mutual trust is important to ensure that the information

passed along supply chains is accurate and not distorted. MSIs are particularly useful when different actors come together and use the exchange as a learning space and can thus exchange ideas at eye level. The "Schallmauer effect" could be overcome through the membership of trade unions in MSIs. However, MSIs should also establish binding force.

### 5.1. Limitations

The limitations to which this research is subject result from the sample selection and the openness of the questionnaire. On the one hand, this had the advantage that aspects were included in the results that were new and had not been constructed on the basis of the theoretical foundations. On the other hand, the openness has a limiting effect on the generalizability of the results. In addition, qualitative research is associated with the possible influence of subjectivity.

### 5.2. Outlook

This work has shown that it could be worthwhile not only to look at the individual transparency technologies in more detail, but also to consider the interaction of the technologies in particular. It also seems useful to conduct further interviews with different actors in order to consolidate the results of the respective perspectives. In doing so, it would be especially important to set a new focus on actors who can accurately reflect the perspective of production sites.

In this regard, it is considered particularly relevant to include the Schallmauer effect in future scientific research on textile industry supply chains, as this is not yet considered in research despite its system-wide relevance. To this end, networks within the textile industry and eye-to-eye dialog need to be strengthened. Also, the research question could be investigated not from the perspective of a trading company, but from the perspective of the workers. This assumes that workers can more easily file complaints or demand compensation if they have information about the supply chain.

**Author Contributions:** Conceptualization, A.C. and C.H.; data curation, C.H.; formal analysis, C.H.; investigation, C.H.; methodology, A.C.; supervision, A.C.; validation, A.C.; writing—original draft, C.H.; writing—review and editing, A.C. All authors have read and agreed to the published version of the manuscript.

**Funding:** The authors would like to thank the Federal Ministry for Economic Cooperation and Development and German Academic Exchange Service for the financial support of this research project (grant number 57504899).

**Data Availability Statement:** Data available on request due to restrictions, e.g., privacy or ethical.

**Conflicts of Interest:** The authors declare no conflict of interest.

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
