# Peer review of "The Influence of Technologies in Increasing Transparency in Textile Supply Chains"

_logistics, 2023_

Round 1

Reviewer 1 Report

Authors have studied the implementation of technologies for increasing the transparency in the textile supply chains. The authors have written the manuscript well, however it is bit lengthy. There is scope of improvement in methods, discussion and conclusions. Reduce the length of manuscript. References are not cited properly. Please see the attached file for comments. 

There are syntax and grammatical errors. Please check it in the manuscript

Author Response

Answers given below each of the reviewer’s comments.

Authors have studied the implementation of technologies for increasing the transparency in the textile supply chains. The authors have written the manuscript well, however it is bit lengthy. There is scope of improvement in methods, discussion and conclusions. Reduce the length of manuscript. References are not cited properly. Please see the attached file for comments. 

Thank you very much for the insightful and kind remarks about our work. Overall, we have shortened the text from 12.474 to 9.209 words to provide a more comprehensive and cohesive overview of our research and its relevance. The citation style of the references was properly adjusted.

Methodology: more detail about what method was used, why this method was choosen, and how the study was conducted, how the data were analyzed, also sample size adequacy in order to present the key findings.

You can elaborate step by step process involved in qualitative interview.

We have provided a more detailed description of the methodology. Lines 113 to 123 provide a more structured overview of the procedure followed to obtain the insights presented in the manuscript. Additionally, lines 86 to 90 provide an explanation of the NGO categories chosen for the research.

Discussion: should be improved better underlining theoretical and practical implications. More associated literature must be added to compare and contrast the key findings with the existing studies. Conclusions should focus mainly on the paper originality.

Conclusions: Provide conclusion of the study – need not give materials and methods

Both the discussion and the conclusions were completely adjusted to provide a better overview of the relevance of the research

Reviewer 2 Report

The primary focus of this article revolves around the negative repercussions resulting from the widespread adoption of the fast fashion strategy by numerous companies, leading to a substantial increase in global textile waste. The complex nature of textile supply chains poses significant challenges in accurately assessing the magnitude of the problem and identifying unethical practices. In order to tackle this issue, the article examines and presents the potential benefits of incorporating technology to improve transparency within textile supply chains. Overall, the paper is well written and presents intriguing findings. However, it is important to acknowledge that there are a few minor limitations that could be addressed to further enhance the quality of this research.

I have listed my concerns below.

·         Section: Abstract An Abstract encompasses key elements such as research objectives, research methodologies, research contents, and quantifiable research outcomes. Failure to address these essential aspects would undermine the effective demonstration of the manuscript's innovation and necessity. By providing this categorical information, readers can readily grasp the efficiency of your research work.

·         Section: Materials and Methods: The novelties of the proposed research is not obvious compared to the existing studies. Therefore, authors are advised to provide a comparison table clearly mentioning the specific contributions of this research. This table should offer a concise yet comprehensive analysis and summary of the existing research, facilitating a clear and tangible understanding of how this study contributes to the existing literature. By incorporating such a comparative analysis, the contribution of this research to the field would be greatly enhanced, while also providing readers with a streamlined and accessible means of comprehending the study's significance.

·      Properly quote the references throughout the manuscript. See following line numbers of the manuscript: 173, 558, 574, 646,699,749,773.

 Minor editing of English language required.

Author Response

Answers given below each of the reviewer’s comments.

The primary focus of this article revolves around the negative repercussions resulting from the widespread adoption of the fast fashion strategy by numerous companies, leading to a substantial increase in global textile waste. The complex nature of textile supply chains poses significant challenges in accurately assessing the magnitude of the problem and identifying unethical practices. In order to tackle this issue, the article examines and presents the potential benefits of incorporating technology to improve transparency within textile supply chains. Overall, the paper is well written and presents intriguing findings. However, it is important to acknowledge that there are a few minor limitations that could be addressed to further enhance the quality of this research.

I have listed my concerns below.

Thank you very much for the insightful and kind remarks about our work.

  • Section: Abstract An Abstract encompasses key elements such as research objectives, research methodologies, research contents, and quantifiable research outcomes. Failure to address these essential aspects would undermine the effective demonstration of the manuscript's innovation and necessity. By providing this categorical information, readers can readily grasp the efficiency of your research work.

The abstract has been completely adjusted to describe the relevance and nature of the research in a more comprehensive and cohesive manner.

  • Section: Materials and Methods: The novelties of the proposed research is not obvious compared to the existing studies. Therefore, authors are advised to provide a comparison table clearly mentioning the specific contributions of this research. This table should offer a concise yet comprehensive analysis and summary of the existing research, facilitating a clear and tangible understanding of how this study contributes to the existing literature. By incorporating such a comparative analysis, the contribution of this research to the field would be greatly enhanced, while also providing readers with a streamlined and accessible means of comprehending the study's significance.

We did not add the proposed comparison table, since a bibliographic literature analysis was not the aim of the study and would not support the results in our point of view. Nonetheless, we understand that the significance of our research has not been well highlighted in the submitted manuscript. Thus, we have described in a more precise manner the current research in the topic and added key publications in the field.

We hope that by doing this, the current research on the topic is properly addressed and suffices for the purpose of this manuscript.

  •     Properly quote the references throughout the manuscript. See following line numbers of the manuscript: 173, 558, 574, 646,699,749,773

Thank you very much for the remarks. We have adjusted the references to the proper citation style.

Reviewer 3 Report

Authors have done nice work. However, comments given below could be addressed.

Have you done the literature review through PRISMA methodology? If not why?

You can elaborate step by step process involved in qualitative interview.

Which category NGO you have chosen for interview (operational and advocacy). Justify

Author Response

Answers given below each of the reviewer’s comments.

Authors have done nice work. However, comments given below could be addressed.

Thank you very much for the insightful and kind remarks about our work.

Have you done the literature review through PRISMA methodology? If not why?
We did not use PRISMA Methodology since the main research focus was on conducting the interviews. Consequently, we draw the main insights from the interviews conducted. We completely understand that by comparing insights from the literature to the interviewees’ responses the reader was misguided to believe that a comprehensive systematic literature review was conducted. We have adjusted the text of the manuscript so that it properly reflects the nature of the conducted research.

You can elaborate step by step process involved in qualitative interview.
We have provided a more detailed description of the methodology. Lines 113 to 123 provide a more structured overview of the procedure followed to obtain the insights presented in the manuscript.

Which category NGO you have chosen for interview (operational and advocacy). Justify
We have added a justification that can be found in lines 86 to 90 of the revised manuscript.

Reviewer 4 Report

Thank you for the opportunity to read a manuscript “THE INFLUENCE OF TECHNOLOGIES IN INCREASING TRANSPARENCY IN TEXTILE SUPPLY CHAINS”. I would like to congratulate author(s) for the performed study. The article is interesting and raises important contemporary issues, however, there are several shortcomings that, in my opinion, author(s) should necessarily address.

1) The abstract should be an objective representation of the article, focus more on the relevance and novelty of the work. There is lack highlighted purpose of the paper. Also, please, describe briefly the main methods applied and then present results.

2) The introduction part lacks the arguments in a solid and critical way why authors are seeking to research technologies. The authors could be more convincing that the subject is crucial, in what elements and why. The current state of the research field should be reviewed carefully and key publications cited. Moreover, the authors should clearly state the main aim of an article.

3) The Methodology section should be presented in more detail about what method was used, why this method was choosen, and how the study was conducted, how the data were analyzed, also sample size adequacy in order to present the key findings.

4) The discussion section should be improved better underlining theoretical and practical implications. More associated literature must be added to compare and contrast the key findings with the existing studies. Conclusions should focus mainly on the paper originality.

5) The text (i.e., lines 173, 558, 574, 646, 699, 749, 773) must be reviewed and corrections made. Image or figure?

6) Citation of authors as well as list of references have to be made according to journals requirements.

I hope the authors should re-think what the contribution of the manuscript is, who should use these results and for what purpose.

Good luck with the revision!

Author Response

Answers given below each of the reviewer’s comments.

Thank you for the opportunity to read a manuscript “THE INFLUENCE OF TECHNOLOGIES IN INCREASING TRANSPARENCY IN TEXTILE SUPPLY CHAINS”. I would like to congratulate author(s) for the performed study. The article is interesting and raises important contemporary issues, however, there are several shortcomings that, in my opinion, author(s) should necessarily address.

Thank you very much for the insightful and kind remarks about our work.

  • The abstract should be an objective representation of the article, focus more on the relevance and novelty of the work. There is lack highlighted purpose of the paper. Also, please, describe briefly the main methods applied and then present results.

The abstract has been completely adjusted to describe the relevance and nature of the research in a more comprehensive and cohesive manner.

  • The introduction part lacks the arguments in a solid and critical way why authors are seeking to research technologies. The authors could be more convincing that the subject is crucial, in what elements and why. The current state of the research field should be reviewed carefully and key publications cited. Moreover, the authors should clearly state the main aim of an article.

As suggested, we have changed the introduction completely and focused on the main implications or our work for research and practice. Furthermore, we have highlighted in a more precise manner the current research in the topic and added key publications in the field. Moreover, the main objective of the research was described in more detail in the adjusted manuscript (lines 61 to 68).

  • The Methodology section should be presented in more detail about what method was used, why this method was choosen, and how the study was conducted, how the data were analyzed, also sample size adequacy in order to present the key findings.

We have provided a more detailed description of the methodology. Lines 113 to 123 provide a more structured overview of the procedure followed to obtain the insights presented in the manuscript.

  • The discussion section should be improved better underlining theoretical and practical implications. More associated literature must be added to compare and contrast the key findings with the existing studies. Conclusions should focus mainly on the paper originality.

Both the discussion and the conclusions were completely adjusted to provide a better overview of the relevance of the research. Additionally, more supporting literature was included.

  • The text (i.e., lines 173, 558, 574, 646, 699, 749, 773) must be reviewed and corrections made. Image or figure?

We have adjusted the references to the proper citation style and renamed the figures.

  • Citation of authors as well as list of references have to be made according to journals requirements.

This was properly adjusted following the APA 7th version citation style.

I hope the authors should re-think what the contribution of the manuscript is, who should use these results and for what purpose.

Thank you very much. In alignment with your suggestions, we adjusted the manuscript to be more concise and cohesive for both practitioners and researchers.

Round 2

Reviewer 3 Report

Authors have addressed the response nicely.